# The Interaction Effect of Birth Spacing and Maternal Healthcare Services on Child Mortality in Pakistan

**DOI:** 10.3390/children10040710

**Published:** 2023-04-12

**Authors:** Muhammad Farhan Asif, Saima Ishtiaq, Nishat Ishfaq Abbasi, Iffat Tahir, Ghulam Abid, Zohra S. Lassi

**Affiliations:** 1Department of Business Administration, ILMA University, Main Campus Korangi Creek, Karachi 75190, Pakistan; 2Department of Statistics, Kohsar University Murree, Murree 47150, Pakistan; 3Department of Botany, Kohsar University Murree, Murree 47150, Pakistan; 4Department of Business Studies, Kinnaird College for Women, Lahore 54000, Pakistan; 5Robinson Research Institute, Adelaide Medical School, University of Adelaide, Adelaide, SA 5000, Australia; 6School of Public Health, Faculty of Health and Medical Sciences, University of Adelaide, Adelaide, SA 5000, Australia

**Keywords:** birth spacing, MHCS, child mortality, Pakistan

## Abstract

There is widespread agreement that improved health should be regarded as a means and an end in the context of the development process. The health of the populace and the equitable provision of healthcare are two indicators of a society’s level of development. A variety of factors influences child mortality. This study investigated the causes of child death and the interaction effect of birth spacing (B.S.) and maternal health care services (MHCS) on child mortality. Using SPSS version 20, we used the Pakistan Demographic and Health Survey (PDHS) 2017–2018 data set to investigate the associated factors of child mortality and the moderating influence of birth spacing using binary logistic regression. The outcome variable is categorical with two categories. The findings indicated that the risk of infant death decreased with adequate B.S. between two pregnancies and access to maternal health care services. Birth spacing was found to moderate the link between access to maternal health care services (MHCS) and child mortality. Our research leads us to conclude that the amount of time between children’s births significantly reduces infant mortality. When the birth spacing is at least 33 months, the relationship between maternal health care services and child mortality becomes more evident and negative.

## 1. Introduction

The right to health is the most important part of the Human Rights Declaration of 1948 [1]. Improving people’s health should be seen as a moral obligation for decision-makers at all levels of government around the globe, including at the international, national, and local levels. Every child has the right to a better life and should be regarded as an asset to their nation as they represent the country’s future human capital. Improved chances of survival before the age of five can lead to a better life in the long run. Therefore, examining more closely whether a child can survive those delicate and important early years of life is vital.

Pakistan is one of the less developed nations with poor health conditions due to its low ranking of 154th in a list of 195 countries in terms of both the availability and quality of medical services [2]. The fact that the country was placed 36th out of 228 countries in terms of child mortality in 2018 is a sign of the deteriorating state of child health in that country. Even though Pakistan has improved child mortality from 141 per 1000 live births in 1990 to 67.2 per 1000 live births in 2019 [3], it is still higher than many developed nations and many less developed nations. Therefore, it is necessary to conduct a more in-depth analysis of the problem of child mortality within the context of Pakistan.

Community-level factors need to be included in the analysis of child mortality to determine which community ecosystems and social structures pose health risks. These variables include immunization, poverty level, the percentage of educated women in the community or literacy rate, ethnicity, the accessibility of healthcare services, the availability of clean water, the community’s hygienic conditions, the cost of healthcare services, social or cultural norms regarding healthcare, residence (urban/rural), etc. [4].

Parental resources significantly impact child survival and mortality. Fathers with a higher education are more likely to provide for their families financially, increasing the likelihood of their children surviving adulthood. If a mother uses excellent preventive and curative health care techniques, her education may boost her productivity in childcare [3,4,5]. Women who have completed their educations might also choose to space out their pregnancies and have a smaller family size overall [6,7,8,9,10,11].

In emerging nations, infant and childhood mortality has been linked to a number of socioeconomic factors. However, depending on a country’s socioeconomic development, social and demographic factors influence infant mortality differently [12]. Park and Mercado (2015) [13] stated that the level of development in a country is affected by differences in economic growth based on income. There is evidence that economic growth influences children’s survival and mortality. Health care factors largely determine neonatal mortality in addition to socio-economic ones. The health of the baby is significantly impacted by women’s health-seeking behaviour, particularly during pregnancy and childbirth [14]. Increased antenatal care visits have been associated with reduced neonatal mortality, possibly because they enhance the mothers’ and babies’ likelihood of receiving and benefiting from health care. The factors of the place of delivery and the presence of medical help during labour and delivery both significantly determine the infant mortality rate. Having a baby in a hospital is linked to a lower infant mortality rate [15].

Women’s empowerment is critical for a country’s economic and social growth and for a child’s health [16]. Poor child health and women’s empowerment are serious public health issues. These disproportionately affect low- and middle-income countries (LMICs) [17]. The involvement of mothers in decision-making has a positive impact on reducing the death of children. Compared to women, men typically choose a large family size and require less contraception than women [18]. Therefore, empowering and equipping women to participate in decisions is vital [19,20].

Individuals with a low socioeconomic status also face difficulty accessing healthcare facilities, and this remains constant in poor nations. According to some studies, access to health care may be classified into five dimensions: cost, accessibility, availability, accommodation, and acceptability [21]. The population’s geographic accessibility to health facilities has not been thoroughly examined, particularly in terms of distance to health facilities. It has been demonstrated that the amount of time and distance required to reach a facility contributes to a declining trend in the utilization of primary health care services [22].

Additionally, mass media exposure is crucial for enhancing maternal and child health. Information on the need to vaccinate children and provide adequate health care is disseminated by various shows broadcast on television. Mothers can take better care of their children due to this awareness [4].

Both the time between births and the availability of maternal healthcare services are key factors in determining the likelihood of child mortality and health [23]. Births spaced out well, between 2 and 4 years, increasing the chances of survival for babies by 2.4 times and for children by 2.9 times [24]. In addition, infants born small for their gestational age have a higher risk of passing away during the neonatal period than children of an average size at birth [23]. The myriad of socioeconomic, demographic, and biological factors are linked to the tragically high child death rate. This study analyses the interactional effect of MHCS and B.S. on child mortality in Pakistan. The country was chosen because it is located in south Asia. Previous investigations, particularly those conducted in Pakistan, have never investigated the precise relationship. To accomplish this goal, we analysed the cross-sectional data gathered as part of the PDHS 2017–2018.

## 2. Methods

### 2.1. Data Source

The dataset from the PDHS 2017–2018 was used for analysis. The Pakistan Bureau of Statistics was decided that a sample size of 16,240 homes, of which 7980 were located in urban areas and 8260 were located in rural regions, would provide sufficient accuracy for the study variables. The procedure for collecting samples consisted of two parts. At the first stage of sampling, primary sample units totalling 580 were selected in accordance with a method known as systematic selection. At the second stage of the sampling procedure, a random selection of 28 households was carried out inside each cluster by employing a systematic sampling method based on the same probability [24]. Approximately 8274 women reported the information for all the study variables. To erase missing data, we used the list-wise deletion procedure. If even a single value in a record was missing, the whole record was skipped in this approach. After excluding the individuals with missing data, the remaining 8274 women’s records were evaluated.

### 2.2. Measurement

CM = Child mortality is the outcome variable of our study. This variable is divided into two categories: household had experienced child mortality coded as 1 and otherwise coded as 0.

M.E.S. = Mothers’ employment status is classified into two categories: 1 if they are currently employed and 0 if they are currently unemployed.

MPDM = Participation of mothers in significant household purchase decisions has been used as a proxy for mothers’ empowerment. It was classified as 0 if the mother has not taken part in the decision-making process and coded as 1 if the mother has taken part in the decision.

M.E.D. = The education of the mother was classified into two categories, coded as 2 if the mother has completed at least ten years of schooling, while it was marked as 1 if the mother has not attained ten years of schooling (no education and primary education).

W.S.H. = Utilizing information on the household’s assets and residence attributes, the W.S.H. was determined. It is divided into five quintiles, from the richest to the poorest. If mothers belong to the richest quintile (richest and richer quintile), they were coded as 2; if they belong to the poorest quintile (poorest, poorer, and middle quintiles), they were recorded as 1.

F.E.D. = There are two groups delineated by the father’s level of education. If the father has completed at least ten years of schooling (secondary school and higher), then it was coded as 2; if the father has not completed ten years of schooling (no education and elementary school), it was coded as 1.

E.M.M. = As a stand-in for exposure to mass media, the availability of television (T.V.) in homes was considered. If the family has a television, the value was 1; otherwise, it was 0.

B.S. = There have been two distinct categories established for birth intervals. Two births separated by more than 33 months was coded as 2, whereas fewer than 33 months was coded as 1.

MHCS = If the mother had availed all these services (minimum 4 antenatal visits, skilled birth attendance, and postnatal visit within 42 days of delivery), this was considered more accessible and coded as 2; if they were not availed all services, this was considered less accessible and coded as 1.

A.T.F. = There are two categories for toilet facility access: 1 for households with access to a toilet facility and 0 otherwise.

ACDW = The degree to which a household has access to clean drinking water is ranked on a scale from 0 (indicating that the household does not have access to clean drinking water) to 1 (indicating that the household does have access to clean drinking water).

UMNFP = UMNFP has been classified into two groups. If women do not have UMNFP, the value was written as 0 (for spacing and limiting), and if women do have UMNFP, the value was coded as 1.

To examine the association between different predictors and child mortality, the following model used was:

CM = f(M.E.S., MEMP, M.E.D., W.S.H., F.E.D., E.M.M., B.S., MHCS, A.T.F., ACDW, UMNFP, MHCS*B.S.)

The descriptive statistics revealed the frequencies and proportions of several participants’ characteristics. The categorical outcome variable showed whether or not there had been child mortalities in the home. When the result variable is a categorical variable with two categories, binary logistic regression was utilized to ascertain the factors that contributed to the death of the children. All analyses were conducted using version 20 of SPSS.

## 3. Results

Table 1 shows the statistics that describe the variables of the study. Approximately eight percent (7.8%) of the women who responded overall had personal experience with child mortality. Around two-thirds of all women (63.7%) and approximately one-third of the husbands of those women (39.9%) did not have an education higher than secondary school. More than half of all women did not have any form of empowerment (56.4%), were not exposed to any form of media (57.1%), and the vast majority of women (88.3%) did not have jobs. A total of 64% of the families were considered low-income; over two-thirds did not have clean drinking water (64.8%), whereas more than three-quarters had A.T.F. (83.9%). Approximately two-thirds of the mothers gave birth to their children at intervals less than 33 months apart (69.2%). One-fifth of the mothers (20.9%) had a UMNFP, while nearly half of the women (49.1%) assessed healthcare facilities as more accessible.

Table 2 explains that the mother’s participation in decision-making (OR = 0.948), mother’s education (OR = 0.685), wealth status of the household (OR = 0.754), father’s education (OR = 0.971), and E.M.M. (OR = 0.904) were negatively linked with child mortality. Similarly, the likelihood of birth spacing (OR = 0.558), and access to a toilet (OR = 0.843), drinking water (OR = 0.799), and maternal health care services (OR = 0.728) were inversely and significantly associated with child mortality. Women with at least ten years of schooling, who are empowered, have access to mass media, have sufficient birth spacing, and have educated husbands had a lower risk of child mortality. The probability of this happening was also lower in the case of wealthy homes with access to protected or clean water and toilet facilities. The UMNFP (OR = 1.102) was positively linked with child mortality.

In other words, women without UMNFP have a decreased risk of child mortality. In the meantime, it was found that maternal health care services had a negative and insignificant impact on the mortality rate of children. The spacing between births as an important contributing factor to reducing the high mortality rate. In this regard, we used an interaction term of birth spacing with maternal health care services.

Child mortality was considerably decreased (OR = 0.801) by the interaction impact of birth spacing and MHCS. The interaction effects of birth spacing and MHCS on child mortality were further illustrated using slope analysis (Figure 1). High birth spacing had a greater negative slope of the curve relating maternal health care services to child mortality than low birth spacing. Increasing the time between births is suggested to improve the relationship between MHCS and child mortality. The influence of MHCS on child mortality became more prominent as birth spacing increased.

## 4. Discussion

According to the findings of the study, factors such as the mother’s education, participation in decision-making, household wealth, exposure to mass media, a father’s education, the availability of toilet facilities, access to clean drinking water, and the amount of time between births all had a negative and significant impact on the risk of a child dying. On the other hand, the mother’s employment status and UMNFP positively affected child mortality. Education for women can benefit not just women but their families and society as a whole. The education of mothers is fundamental to improving child health and child survival. The higher the mother’s degree of education, the greater her understanding of risk factors connected with child mortality, such as early marriage control to prevent adolescent pregnancy. The father’s education minimizes child mortality through his increased salary and productivity, which improves the family’s financial conditions and maternal, child, and family health through efficient consumption habits. Child mortality is highly associated with both maternal and paternal education levels.

The likelihood of child mortality was higher for employed women. Employed women are believed to be overwhelmed with domestic duties, and their low-paying jobs limit their access to health care [25]. The effect of women’s employment positions on the utilization of maternity and child health services has varied among studies. Researchers have found a negative correlation between healthcare services and women’s employment status; employed women suffer time restrictions that reduce their utilization of healthcare services. Our findings are consistent with previous research [26,27].

Women with more years of schooling are typically more independent and are expected to participate more in household decision-making. Women with more autonomy, bargaining power, and empowerment are more likely to use contraceptives more frequently and have lower fertility rates [19]. Female autonomy in decision-making reduces unintended pregnancies by as much as 57% [28]. Women’s influence over home expenditures increases spending on children’s health, education, and nutrition. Individuals have also noticed that if both the husband and wife earn money for the family, most of the women’s money goes towards the health and nutrition of their children [19,29]. Maitra (2004) [30] believes that when women have more freedom, there is a large decline in the number of children who die before their first birthday. This is because women obtain better prenatal care and have more chances to give birth in a hospital.

To reduce child mortality, it is crucial to increase awareness among women. Mass media plays a major role and provides access to critical and practical information about child immunization and the right health care that should be offered to children. This understanding enables mothers to adequately care for their infants, reducing child mortality [31]. The education of the father is negatively linked with child mortality. A person with more education is more likely to have a higher income, which is important for the family’s financial stability and access to health care [32]. A male who has received formal education has a greater chance of understanding and appreciating the significance of maintaining optimal child health. Child mortality rates are lower in more financially secure households. People in wealthier households live better and have better access to health care. Especially in more affluent communities, this may help lower the infant death rate. However, households with higher socioeconomic levels are more likely to have access to healthcare facilities and basic sanitation amenities like clean water and flush toilets [33].

The prevalence of diarrhoea, one of the leading causes of child mortality, can be reduced by providing children with access to toilets [34]. In 2017, diarrheal illness was responsible for 0.53 million deaths worldwide [35]. It was also the cause of death for one out of every ten children who passed away. Participation in water, sanitation, and hygiene programmes is the most effective way to prevent diarrheal infections [36], which in turn can assist in lowering the mortality rate of children [34].

Additionally, the findings indicated that birth intervals affect infant mortality [37,38]. Children born with birth intervals of fewer than 24 months have a lower chance of surviving than children born after more than 24 months. This was also confirmed by the overall findings, which indicate that the infant mortality rate for children with a shorter birth interval was 11.8% and that for children with a longer birth interval was 4.9% [39].

Child mortality was positively linked with UMNFP. Numerous socio-economic and cultural variables contribute to Pakistan’s high prevalence of unmet family planning needs. Access to contraceptives for women is restricted for a number of reasons, including fear of side effects and sociocultural constraints. Effective public policy interventions and social mobilization with the participation of religious and community leaders can be useful in lowering the percentage of people who do not receive the family planning services they require [40,41].

Children born to healthy mothers are also healthy. Childhood developmental problems can result from the mother’s poor health (malnutrition) [42]. The child is more likely to have health issues or pass away if the mother is not in excellent health throughout the pregnancy and/or the foetus is exposed to teratogen(s), including radiation, infectious agents, hormones, chemical agents, nutritional deficits, and maternal illnesses [31,43].

It is important to note that birth spacing moderated the relationship between maternal healthcare services and infant mortality. The association between maternal health care services and child mortality was strengthened by birth spacing. The reduction in infant mortality caused by access to quality maternal health care increased in magnitude as births are spaced further apart. The World Health Organization recommends a minimum of 33 months of birth spacing between births or 24 months before trying a future pregnancy to decrease the risk of unfavourable maternal and newborn outcomes. The short time between pregnancies makes it hard for the mother to obtain all the nutrients she needs, especially iron and folate, for the subsequent pregnancy. This lack of nutrients is detrimental for the health of both mothers and their children. Similarly, having a short time between pregnancies is linked to having a baby with a small gestational age, giving birth prematurely, having a low birth weight, having a stillbirth, and increasing rates of child and mother mortality [44].

## 5. Recommendation

Our research on Pakistan supports the significance of these key risk factors for infant mortality, namely those connected to the child, the mother, and the household, such as socioeconomic position, environmental conditions, social behaviour, and health services utilization. These results demonstrated the need for future child health programs to take a broader strategy to significantly reduce child mortality in Pakistan, particularly in rural and underdeveloped areas. Improvements in the accessibility of formal health services, special focus on children during their first year of life, emphasis on teaching entire families about birth spacing, and efforts to make antenatal care, institutional delivery, and postnatal services more accessible to mothers are all necessary components of such programs. They also need to continue to improve households’ economic standing.

## 6. Limitations and Direction for Further Study

This study examined the moderating effect of birth spacing on the association between maternal healthcare services and child mortality in Pakistan. The current research is limited since we did not determine whether or not other factors affected the hypothesized relationship. It is important to note that several other causes could also affect child mortality, all of which need to be investigated. For instance, a father’s educational background may play a role in the relationship between the household’s wealth status and the child mortality rate. This is due to the fact that a father’s education plays an essential role in the process of raising awareness about healthy and nutritious food, which is beneficial for the health of children. He will be attentive to prevent children from consuming tainted and preserved foods.

## 7. Conclusions

In conclusion, it was determined that birth spacing and maternal health care services are key determinants of child mortality. Compared to low birth spacing, high birth spacing had a stronger negative slope on the curve, indicating a relationship between maternal health care services and child mortality. Increasing the time between births improved the connection between MHCS and child mortality. As birth spacing increased, the effect of MHCS on infant mortality became more pronounced. According to the findings from this study, there is an urgent need to address the social variables influencing maternal and infant mortality in Pakistan. Investments in females’ education are vital due to the importance of a mother’s education for enhancing child survival. The Ministry of Health and the provinces must work together to ensure that healthcare messages about child and maternal health include other areas, such as employment, schooling, empowerment, birth spacing, and preventing children from getting married too young. The formal and informal education of women should be a priority for governments so that they may contribute more effectively to both the advancement of the country and their own personal lives. The country must utilize the mass media’s private and public sectors to raise knowledge of the benefits of birth spacing and family planning, particularly for women’s and children’s health.

## Figures and Tables

**Figure 1 children-10-00710-f001:**
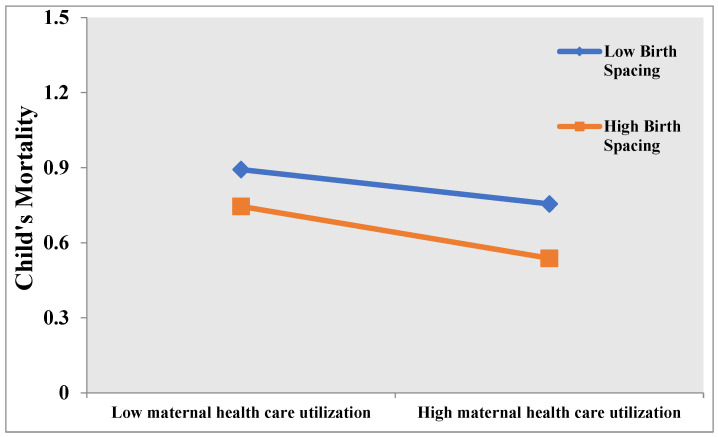
Interaction Effect of Birth Spacing and MHCS on Child Mortality.

**Table 1 children-10-00710-t001:** Descriptive statistics of different characteristics of respondents.

Descriptive		Frequency 8274	Percent (%)
Child’s living status	Alive	7632	92.2
	Not alive	642	7.8
Mother’s employment status	Currently unemployed	7305	88.3
	Currently employed	969	11.7
Mother participates in decision making	Less empowered	4672	56.4.
	More empowered	3602	43.6
Mother’s education	Less than ten years of schooling	5271	63.7
	At least ten years of schooling	3003	36.3
Household wealth index	Poorest	5306	64.1
	Richest	2968	35.9
Father’s education	Less than ten years of schooling	3302	39.9
	At least ten years of schooling	4972	60.1
Exposure to mass media	No exposure	3395	41.0
	Exposure	4878	59.0
Birth spacing	Less than 33 Months	5726	69.2
	At least 33 Months	2548	38.8
Maternal health care services	Less accessible	4210	50.9
	More accessible	4064	49.1
Access to toilet facilities	No access	1332	16.1
	Access	6942	83.9
Access to clean drinking water	No	5362	64.8
	Yes	2912	35.2
Unmet need for family planning	No	6611	79.9
	Yes	1663	20.1

**Table 2 children-10-00710-t002:** The relationship between socio-economic determinants and child mortality: moderating role of birth spacing.

Dependent Variable: Child Mortality				
Independent Variables		Β	Sig.	Odd Ratio (OR)
Mother’s Employment Status	Currently unemployed	Ref		
	Currently employed	0.248	0.000	1.299
Mother Participates in Decision-Making	No	Ref		
	Yes	−0.068	0.051	0.981
Mother’s Education	Less than ten years of schooling	Ref		
	At least ten years of schooling	−0.351	0.000	0.685
Wealth Status of Household	Poorest	Ref		
	Richest	−0.319	0.000	0.754
Father’s Education	Less than ten years of schooling	Ref		
	At least ten years of schooling	−0.092	0.083	0.971
Exposure to Mass Media	No	Ref		
	Yes	−0.103	0.013	0.904
Birth Spacing	Less than 33 Months	Ref		
	At least 33 Months	−0.572	0.001	0.558
Maternal Health Care Services	Less utilization	Ref		
	More utilization	−0.266	0.273	0.728
Access to Toilet Facilities	No	Ref		
	Yes	−0.179	0.000	0.843
Access to Clean Drinking Water	No	Ref		
	Yes	−0.214	0.000	0.799
Unmet Need for Family Planning	No	Ref		
	Yes	0.117	0.046	1.102
Birth Spacing * Maternal Health Care Services		−0.122	0.019	0.801

* indicates the interaction term.

## Data Availability

We used the secondary data of the PDHS 2017–2018 available at: https://www.nips.org.pk/study_detail.php?detail=MTgw (accessed on 24 September 2022).

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
