# Peer review of "The Interaction Effect of Birth Spacing and Maternal Healthcare Services on Child Mortality in Pakistan"

_children, 2023, doi:10.3390/children10040710_

Round 1

Reviewer 1 Report

Dear Authors, 

I would like to congratulate you for the interesting paper that you have provided to the journal.

After reading your manuscript, I would like to emphasize the following:

- the abstract is quite well structured, presenting a clear objective of the paper, but also the methodology and also the final message of the article. Even so, please pay attention to the language - some sentences are incomplete. For instance, at line 25 - ”While the chance of child death increases with the unmet need for family planning (UMNFP) and women's work” 

- the introduction is also well structured, but also please pay attention to the language. In some cases, the sentences seem to be translated with Google translate, like at line 34 - ”The current hot topic is the health of countries, with the logic being that a healthy population indicates a healthy country”. 

Also at line 68-69 - ”Since literature shows that economic growth also affects child survival, child mortality also affects.” It does not sound academic at all. 

I appreciate that the paper has clear objectives and also pints out the added value of the study to the existing body of literature written on the topic.

Concerning the methodology, in my point of view the research method should be better explained. For someone that is not familiar with binary logistic regression, I think that a short description of the method, why it is proper for their endeavor and what does it highlights is essential in order to better frame the methodological section. 

Considering the rest of the paper, I find it very interesting and necessary considering current challenges in terms of child mortality and it determinants worldwide.

Thank You! 

Author Response

I would like to congratulate you for the interesting paper that you have provided to the journal.

Response: Thank you for your appreciation.

The abstract is quite well structured, presenting a clear objective of the paper, but also the methodology and also the final message of the article. Even so, please pay attention to the language - some sentences are incomplete. For instance, at line 25 - ”While the chance of child death increases with the unmet need for family planning (UMNFP) and women's work” 

Response: Thank you for your valuable comment. Comment incorporated

The introduction is also well structured, but also please pay attention to the language. In some cases, the sentences seem to be translated with Google translate, like at line 34 - ”The current hot topic is the health of countries, with the logic being that a healthy population indicates a healthy country”. 

Response: Comments incorporated. Avoid the translation from google translator.

Also at line 68-69 - ”Since literature shows that economic growth also affects child survival, child mortality also affects.” It does not sound academic at all. 

Response: Comments incorporated. Please see the lines of 64-66.

I appreciate that the paper has clear objectives and also pints out the added value of the study to the existing body of literature written on the topic.

Response: Thank you for your appreciation. Comment incorporated. Please see the lines of 97-102.

Concerning the methodology, in my point of view the research method should be better explained. For someone that is not familiar with binary logistic regression, I think that a short description of the method, why it is proper for their endeavor and what does it highlights is essential in order to better frame the methodological section. 

Response: Thank you for your valuable suggestion. Comment incorporated, and we give a brief description of binary logistic regression. Please see the lines of 163-166.

Considering the rest of the paper, I find it very interesting and necessary considering current challenges in terms of child mortality and it determinants worldwide.

Response: Thank you for your appreciation.

Reviewer 2 Report

The manuscript tried to investigate the impact of maternal healthcare services and birth spacing on the mortality rate of child. In this work, Sociodemographic study was used for estimating the corresponding biomarker probability. These results may provide a potential approach as a case study of specific region.

I think the abstract section needs more effort. For instance, in the introduction of the abstract, you mentioned all the general terms of the widespread agreement that improved health, without focusing on your manuscript novel objectives. You should mention why your study compared to the higher number sources. The methods section needs some modification in the used techniques in more details. What is the novel information that presented from the sentence “the risk of child mortality drops along with a woman's level of education and empowerment, as well as her husband's level of education..” in the abstract? Thus, you should have conclusion with the novelty of your findings. “the relationship between maternal health care services and child mortality becomes more evident and positive” in the abstract without mentioning the novelty of these observations? In the introduction section, you should mention the recent studies compared to your used marker characteristics. Without a thorough literature review, referees and editors are much less likely to accept that the research is sufficiently topical or original. Furthermore, by citing recent articles from currently active researchers from internationally available journals your research is much more likely to attract attention and to be read and cited. You should aim to improve the focus of both the Introduction and Discussion sections upon the latest research. In what is a very active area of research too many of the articles cited are more than 5 years old and too few less than 3 years old (2020-2022). Greater emphasis should be made of the most recent research from authoritative international Food science journals. There are several grammatical and typo errors. English editing is suggested. Punctuation related errors are there, without any punctuation. How could you discuss the changes without full anthropometric analyses for the mother and children? I think the more efforts should be for the biomarker chemical characterization like blood and/or urinary analyses.

In the discussion: Generally, this section needs to be supplemented with more other literature review papers. Please add the mechanism of action. I think you should discuss the innovations and more clarifying why your study in this section in more details.

In conclusion: More details about your observations by showing some of this valuable study's significations point by point should be clarified.

The optimal Conclusion should include:
• A summary of your key findings.
• A highlight of your hypothesis, new concepts, and innovations.
• A summary of key improvements compared to findings in the literature [provide a couple of references to indicate key improvements].
• Your vision for future work.

The Tables and figure values needs the SD, median, and quartiles.   

Author Response

I think the abstract section needs more effort. For instance, in the introduction of the abstract, you mentioned all the general terms of the widespread agreement that improved health, without focusing on your manuscript novel objectives. You should mention why your study compared to the higher number sources.

Response: Thank you for the suggestion. Comment incorporated. The interactional effect of maternal health care services and birth spacing on child mortality has never been seen in previous studies in the context of Pakistan.

The methods section needs some modification in the used techniques in more details.

Response: Comment incorporated and can be seen in method section.

What is the novel information that presented from the sentence “the risk of child mortality drops along with a woman's level of education and empowerment, as well as her husband's level of education..” in the abstract? Thus, you should have conclusion with the novelty of your findings.

Response: Comment incorporated in the last part of the abstract.

In the introduction section, you should mention the recent studies compared to your used marker characteristics. Without a thorough literature review, referees and editors are much less likely to accept that the research is sufficiently topical or original. Furthermore, by citing recent articles from currently active researchers from internationally available journals your research is much more likely to attract attention and to be read and cited. You should aim to improve the focus of both the Introduction and Discussion sections upon the latest research. In what is a very active area of research too many of the articles cited are more than 5 years old and too few less than 3 years old (2020-2022). Greater emphasis should be made of the most recent research from authoritative international Food science journals. 

Response: Thank you for the suggestion. The citation has been updated.

There are several grammatical and typo errors. English editing is suggested. Punctuation related errors are there, without any punctuation.

Response: The whole document is revisited for any grammatical and typo errors. Thank you.

How could you discuss the changes without full anthropometric analyses for the mother and children? I think the more efforts should be for the biomarker chemical characterization like blood and/or urinary analyses.

Response: Thank you for your suggestion. Our study aims to investigate the interactional effect of maternal healthcare services and birth spacing on child mortality in Pakistan. So the biomarker chemical characterization like blood and/or urinary analyses has been checked in future research.

In the discussion: Generally, this section needs to be supplemented with more other literature review papers. Please add the mechanism of action. I think you should discuss the innovations and more clarifying why your study in this section in more details.

Response: Thank you for the suggestion. Comment incorporated and updated papers are cited. The channel between dependent and all independent variables has been discussed in the discussion section.

In conclusion: More details about your observations by showing some of this valuable study's significations point by point should be clarified.

Response: Changes have been made and can be seen in the conclusion part.

Reviewer 3 Report

Well-done on your study. Please see below comment to make your study stronger.

1.) In your introduction you mentioned community-level factors to include variables such as immunization, poverty level, the percentage of educated women in the community or literacy rate, ethnicity, the accessibility of healthcare services, the availability of clean water, the community's hygienic conditions, the cost of healthcare services, social or cultural norms regarding healthcare, residence (urban/rural), etc.

2.) What informed this choice of variables? a study? a conceptual framework? any references?

3.) What is the rationale for your study? you have mentioned a lot of factors associated with child mortality but the buildup to your specific study aim is lacking a "punch line". What gap in the literature is your study aiming to fill? Why do we need another study on the factors associated with child mortality? were the past studies methodologically flawed?

4.) Provide more information on PDHS 2017-18 backed up with a reference.

5.) What is/are the dependent variable(s)?

6.) What are the descriptive Study Variables?

7.) Why is the coding for maternal education different from that of father's education?

8.) The coding for WSH masks the wealth distribution. It would be advisable to classify WSH into richest quintile (richest and richer quintile); middle quintile; and poorest quintile (poorest and poorer quintiles).

9.) How was the statistical analysis conducted? Provide more details.

10.) In the first sentence of the discussion section, state the direction of association e.g low mother's education, employed mothers, low participation in decision-making, etc.

11.) Your discussion on employed women is hard to comprehend. The first sentence in the paragraph (line 217) seem to contradict the rest of the paragraph. Moreso, are you discussing child mortality and employed women or are you discussing women work status and utilization of healthcare services. Please rewrite the entire paragraph to discuss your study finding.

12.) What are the strengths of your study?

Author Response

Well-done on your study. Please see below comment to make your study stronger.

Response: Thank you for your appreciation.

In your introduction you mentioned community-level factors to include variables such as immunization, poverty level, the percentage of educated women in the community or literacy rate, ethnicity, the accessibility of healthcare services, the availability of clean water, the community's hygienic conditions, the cost of healthcare services, social or cultural norms regarding healthcare, residence (urban/rural), etc. What informed this choice of variables? a study? a conceptual framework? any references?

Response: Thank you for your suggestion. The comment has been incorporated and cited the study.

What is the rationale for your study? you have mentioned a lot of factors associated with child mortality but the buildup to your specific study aim is lacking a "punch line". What gap in the literature is your study aiming to fill? Why do we need another study on the factors associated with child mortality? were the past studies methodologically flawed?

Response: Comment has been incorporated. Please see the last five lines of the introduction section.

Provide more information on PDHS 2017-18 backed up with a reference.

Response: Comment has been incorporated. Please see data source section.

What is/are the dependent variable(s)? What are the descriptive Study Variables?

Response: Comment incorporated. The dependent variable of the study is child mortality, and the remaining variables are independent variables.

Why is the coding for maternal education different from that of father's education?

Response: Comment incorporated. Change the coding of the maternal education variable.

The coding for WSH masks the wealth distribution. It would be advisable to classify WSH into richest quintile (richest and richer quintile); middle quintile; and poorest quintile (poorest and poorer quintiles).

Response: In many previously published articles, the household's wealth status has been categorized into two categories. Mothers from the poorest, poorer, and middle quintiles were considered the poorest and if women belonging to the richer and richest quintiles were considered the richest.

How was the statistical analysis conducted? Provide more details

Response: Comments incorporated. Please see lines 163-167.

In the first sentence of the discussion section, state the direction of association e.g low mother's education, employed mothers, low participation in decision-making, etc.

Response: Comments have been incorporated, and rewrite this sentence according to your suggestion.

Your discussion on employed women is hard to comprehend. The first sentence in the paragraph (line 217) seem to contradict the rest of the paragraph. Moreso, are you discussing child mortality and employed women or are you discussing women work status and utilization of healthcare services. Please rewrite the entire paragraph to discuss your study finding.

Response: Thank you for highlighting our mistake. The whole paragraph is rewritten. Please see the lines of 223-229.

What are the strengths of your study?

Response: This study investigates the interactional effect of maternal healthcare services and birth spacing on child mortality in Pakistan. The specific relation has never been seen in previous studies, especially in Pakistan. It is the strength of our study.

Round 2
